# Antiviral Activities of *Streptomyces* KSF 103 Methanolic Extracts against Dengue Virus Type-2

**DOI:** 10.3390/v15081773

**Published:** 2023-08-20

**Authors:** Nurfatihah Zulkifli, Jasmine-Elanie Khairat, Adzzie-Shazleen Azman, Nur-Faralyza Mohd Baharudin, Nurul-Adila Malek, Syafiq-Asnawi Zainal Abidin, Sazaly AbuBakar, Pouya Hassandarvish

**Affiliations:** 1Institute of Biological Sciences, Faculty of Science, Universiti Malaya, Kuala Lumpur 50603, Wilayah Persekutuan Kuala Lumpur, Malaysia; fatihahzulkifli.work@gmail.com (N.Z.); jasmine@um.edu.my (J.-E.K.); nurfaralyza95@gmail.com (N.-F.M.B.); adilamalek@gmail.com (N.-A.M.); 2Tropical Infectious Diseases Research and Education Centre (TIDREC), Universiti Malaya, Level 2, High Impact Research (HIR) Building, Kuala Lumpur 50603, Wilayah Persekutuan Kuala Lumpur, Malaysia; sazaly@um.edu.my; 3School of Science, Monash University Malaysia, Jalan Lagoon Selatan, Bandar Sunway 47500, Selangor, Malaysia; adzzieshazleen.azman@monash.edu; 4Jeffrey Cheah School of Medicine and Health Sciences, Monash University Malaysia, Jalan Lagoon Selatan, Bandar Sunway 47500, Selangor, Malaysia; syafiq.asnawi@monash.edu

**Keywords:** antivirals, arbovirus, dengue virus, *Streptomyces*

## Abstract

Dengue has long been a serious health burden to the global community, especially for those living in the tropics. Despite the availability of vaccines, effective treatment for the infection is still needed and currently remains absent. In the present study, the antiviral properties of the *Streptomyces* sp. KSF 103 methanolic extract (*Streptomyces* KSF 103 ME), which consists of a number of potential antiviral compounds, were investigated against dengue virus serotype 2 (DENV-2). The effects of this extract against DENV-2 replication were determined using the quantitative Real Time-Polymerase Chain Reaction (qRT-PCR). Findings from the study suggested that the *Streptomyces* KSF 103 ME showed maximum inhibitory properties toward the virus during the virus entry stage at concentrations of more than 12.5 µg/mL. Minimal antiviral activities were observed at other virus replication stages; adsorption (42% reduction at 50 µg/mL), post-adsorption (67.6% reduction at 50 µg/mL), prophylactic treatment (68.4% and 87.7% reductions at 50 µg/mL and 25 µg/mL, respectively), and direct virucidal assay (48% and 56.8% reductions at 50 µg/mL and 25 µg/mL, respectively). The *Streptomyces* KSF 103 ME inhibited dengue virus replication with a 50% inhibitory concentration (IC_50_) value of 20.3 µg/mL and an International System of Units (SI) value of 38.9. The *Streptomyces* KSF 103 ME showed potent antiviral properties against dengue virus (DENV) during the entry stage. Further studies will be needed to deduce the antiviral mechanisms of the *Streptomyces* KSF 103 ME against DENV.

## 1. Introduction

DENV is a single-stranded RNA virus that belongs to the genus flavivirus, family *Flaviviridae*. It is the most rapidly spreading arboviral disease in the world, and is mainly found in the tropics, where the mosquito vectors *Aedes aegypti* and *Aedes albopictus* are found [1,2,3]. According to the World Health Organization (WHO), the estimated global population at risk for dengue is about 3.9 billion people. There are four serotypes of DENV: DENV serotype 1 (DENV-1), DENV serotype 2 (DENV-2), DENV serotype 3 (DENV-3), and DENV serotype 4 (DENV-4). The genome consists of three structural proteins, capsid (C), premembrane (prM), and envelope (E) proteins, and seven non-structural proteins, NS1, NS2A, NS2B, NS3, NS4A, NS4B, and NS5 [4]. All four DENVs can cause a wide range of illnesses, from asymptomatic to flu-like illnesses. Although rare, some people may develop severe dengue, which manifests with severe plasma leakage that could result in subsequently several complications, including multiple organ dysfunction [2,4,5]. DENV infects about 400 million individuals annually and causes approximately 40,000 deaths [6,7]. Patients rely mainly on supportive treatment such as the use of analgesics, intravenous fluid, and vital monitoring. In some cases where the disease develops into severe dengue, hospitalization is often needed. Treatment measures that would prevent severe dengue manifestations are thus desperately needed.

To date, there are two vaccines approved by the WHO, namely CYD-TDV and TAK-003. Despite having these vaccines, the former requires seropositive individuals to be effective, while the latter, though serostatus-independent, shows an inconsistent efficacy among dengue serotypes [8,9,10]. Previous studies have shown that the level of viremia in dengue patients corresponds to the severity of the disease [5,11]. The viremia level was higher in secondary dengue infection, which may prolong the disease and increase the chances of manifesting severe dengue symptoms [5,11,12,13,14]. Treatments such as antivirals to reduce the level of viremia, hence, could possibly reduce the severity of dengue. Escalation in research efforts to discover and develop effective antivirals is much needed since the only treatment measure for dengue relies mainly on supportive care, such as the use of analgesics or antipyretics and fluid management. 

Antivirals may act against DENV in several ways, including inhibiting the virus replication cycle, direct inhibition, or prophylactic treatment [1]. Understanding DENV replication cycles is therefore important. Replication cycles of DENV consist of adsorption of the virus to the cell, virus entry into the cell, release of the virus genome, assembly, and release of infectious virus particles into the cell [15]. Various strategies are being explored to hinder the replication of the dengue virus, including targeting the NS3 protease, NS5 polymerase, and virus entry process. The NS3 protease plays a vital role in viral replication, and inhibiting this enzyme could disrupt the virus’s life cycle [10,12,13,14]. Several compounds have demonstrated promise in inhibiting the NS3 protease during preclinical studies [10,12,13,14]. The NS5 polymerase, another crucial enzyme involved in viral replication, is also being targeted as a potential avenue to prevent viral replication [10,16,17]. Numerous compounds have shown inhibitory effects on the NS5 polymerase in laboratory studies. Additionally, researchers are investigating antiviral agents that aim to block the virus’s entry into host cells during the early stages of infection. 

Some compounds have exhibited potential in inhibiting dengue virus entry, but further research is necessary to develop effective treatments [10,18]. It is important to note that although these potential antiviral candidates have shown promise in preclinical studies, they have yet to advance to clinical trials or receive regulatory approval for human use. The development of antivirals involves rigorous testing, including clinical trials, to ensure their safety and effectiveness. A number of antivirals were in use for the treatment of virus infection, and these include ribavirin and acyclovir [10,19], whereas antivirals such as virantmycin and alanosine derived from secondary metabolites of soil microorganisms, particularly *Streptomyces*, have been shown to possess potent antiviral properties [20,21,22]. There are other bioactive compounds derived from *Streptomyces* that were reported to be effective against other viruses. For example, narasin, an ionophore derived from *Streptomyces aureofaciens* has been shown to inhibit DENV replication post-entry at a concentration of less than 1 µM against all DENV serotypes [15].

In a recent study, *Streptomyces* KSF 103 was isolated from soil sediments of a primary forest in Jerantut, Pahang, Malaysia [23]. An ethyl acetate extract of strain KSF 103 possessed potent insecticidal activities against *Aedes* and *Culex* mosquito sp. [23]. Here, the effects of *Streptomyces* KSF 103 ME on DENV-2 replication were investigated. While our study contributes to the understanding of the antiviral potential of *Streptomyces* KSF 103 ME against DENV-2 replication, we acknowledge that this study is limited to in vitro study and propose future research directions to address these limitations and further explore the therapeutic applications of *Streptomyces*-derived compounds in the fight against dengue.

## 2. Materials and Methods

### 2.1. Extraction of Streptomyces sp. KSF 103

*Streptomyces* sp. KSF 103 (Kuala Sat Forest number 103; voucher specimen available at Tropical Infectious Diseases Research and Education Centre) was isolated from Jerantut, Pahang, Malaysia. The extraction of *Streptomyces* KSF 103 ME was performed as previously established and described in our laboratory [10]. The extract was then used for subsequent studies to determine the antiviral activity of the extract against DENV-2. Briefly, a pure culture of *Streptomyces* strain KSF 103 was obtained via streak method on yeast-malt extract (ISP2) media and stored in glycerol suspension (30%, *v*/*v*) [24]. When needed, the glycerol suspension was recovered on ISP2 media and incubated for 5 days at 28 °C as previously described [25,26]. Following that, the bacterial suspension was added to ISP2 broth and incubated at 28 °C for 7–10 days with constant shaking. To avoid any contamination during fermentation due to environmental factors, a flask containing sterile ISP2 broth was incubated with the sample as a control. Then, the bacterial culture suspension was centrifuged at 2500× *g* at 4 °C for 30 min. The supernatant was recovered and freeze-dried. Methanol was added to the freeze-dried residue at a 3:1 ratio (methanol:sample) for 72 h and the liquid was collected via filtration using a filter paper (Whatman). The above step was repeated for ratios 2:1 and 1:1 at the same condition with 24 h intervals. All the methanol-containing compound was then extracted using a rotary vacuum evaporator (Heidolph Laborota 4011, Schwabach, Germany). The crude extract was weighed, and the final *Streptomyces* KSF 103 ME was stored at −20 °C until needed [24]. For the experiments, the extract was thawed at room temperature and dissolved in dimethylsulfoxide (DMSO) (Sigma, Darmstadt, Germany). The extract was serially diluted in Dulbecco’s Minimum Essential Medium (DMEM) (Gibco, Grand Island, NY, USA), and supplemented with 10% fetal bovine serum (FBS, Gibco, Grand Island, NY, USA) into 50 µg/mL, 25 µg/mL, 12.5 µg/mL, 6.25 µg/mL, and 3.125 µg/mL. 

### 2.2. Cell Culture and Virus

Vero cells (African green monkey kidney cells) (ATCC, Manassas, VA, USA) and C6/36 mosquito cells (ATCC, Manassas, VA, USA) were cultured in DMEM, and supplemented with 10% (FBS). The cells were incubated at 37 °C and 28 °C supplemented with 5% and 3% CO_2_ humidified atmosphere, respectively. The Vero cells were used for antiviral screening and cytotoxicity assay, while C6/36 mosquito cells were used for propagation and titration of the virus. The DENV-2 New Guinea C strain (NGC) (ATCC, Manassas, VA, USA) was used in this study. 

### 2.3. Cytotoxicity Assay

Briefly, Vero cells were seeded into 96-well cell culture microplates and incubated overnight at 37 °C with 5% CO_2_. After the incubation, the cell monolayers were treated with different concentrations of *Streptomyces* KSF 103 ME up to 500 µg/mL in duplicate. The treated cells were incubated for four days at 37 °C with 5% CO_2_. After the incubation, MTS reagent (3-(4,5-dimethylthiazol-2-yl)-5-(3-carboxymethoxyphenyl)-2-(4-sulfophenyl)-2H-tetrazolium) (Promega, Madison, WI, USA) was added to each well, and the microplate was incubated at 37 °C for 4 h. To determine the cell viability, the optical density (OD) of the treated cells was measured at 490 nm to 700 nm using TECAN 96-well plate reader (TECAN, Männedorf, Switzerland). Dose–response curves were plotted using Graph Pad Prism Version 5 (Graph Pad Software Inc., San Diego, CA, USA), and the values of 50% cytotoxic concentration (CC_50_) and Maximum Non-Toxic Dose (MNTD) were calculated.

### 2.4. Screening for Inhibitory Activities of Streptomyces KSF 103 ME against DENV-2

A monolayer of Vero cells cultured in a 24-well cell culture microplate, in DMEM, containing 10% FBS (Gibco, Grand Island, NY, USA) was used. The *Streptomyces* KSF 103 ME, at different concentrations, were added to the wells with DENV-2 multiplicity of infection (MOI) ≈ 0.1 [27,28]. DENV-2 control with no treatment and non-infected cells treated with 0.1% DMSO were included in the plate design as comparison. The plate was then incubated at 37 °C with 5% CO_2_ for 96 h. The assay was performed in duplicate for each concentration of the extract. After 4 days, the cell culture supernatant was harvested, aliquoted, and kept at −80 °C until needed for virus RNA extraction and evaluation using qRT-PCR. 

### 2.5. Identifying the Effects of Streptomyces KSF 103 ME on Different Stages of DENV-2 Replication Cycle

Vero cells were seeded into a 24-well cell culture microplate and incubated overnight or until the cells become confluent at 37 °C supplemented with 5% CO_2_. For the anti-adsorption study, the cell culture microplate was pre-chilled at 4 °C for 30 min. The cells were treated with different concentrations of *Streptomyces* KSF 103 ME (50 µg/mL, 25 µg/mL, 12.5 µg/mL, 6.25 µg/mL, and 3.125 µg/mL) and infected with DENV-2 at MOI ≈ 0.1 and incubated at 4 °C for 1 h. The anti-entry study was performed using the confluent Vero cells infected with DENV-2 at MOI ≈ 0.1 and incubated at 4 °C for 1 h for virus adsorption. Then, the cells were treated with different concentrations of *Streptomyces* KSF 103 ME and incubated at 37 °C with 5% CO_2_ for 1 h to allow for the inhibitory activity of the extract against DENV-2 entry. Then, the supernatant was removed, and 200 µL of citrate buffer (citric acid buffer 40 mM, KCl 10 mM, NaCl 135 mM, pH 3.0) was added to the cells and incubated for 1 min to remove unbound DENV-2 particles. For the post-adsorption antiviral assay, the confluent cells were infected with DENV-2 at MOI ≈ 0.1 and incubated at 37 °C with 5% CO_2_ for 1 h. After virus adsorption, the cells were treated with different concentrations of *Streptomyces* KSF 103 ME. Prophylactic treatment was performed by treating the confluent cells with the KSF 103 ME at different concentrations and incubated at 37 °C with 5% CO_2_ for 5 h. The direct virucidal assay was performed using DENV-2 at MOI ≈ 10. DENV-2 was pre-incubated with the different concentrations of *Streptomyces* KSF 103 ME at a 1:1 ratio for 2 h at 37 °C with 5% CO_2_. Then, the treated virus inoculum was diluted 100-fold and added to the confluent cells and incubated at 37 °C with 5% CO_2_ for 1 h to allow adsorption of DENV-2. DMEM with 2% FBS was added, and the plates were incubated at 37 °C with 5% CO_2_ for 4 days. After the incubation, the antiviral activity of the extracts was determined by the reduction in RNA copy numbers using qRT-PCR. All experiments included DENV-2 control with no treatment and non-infected cells treated with 0.1% DMSO in the plate design as comparison.

### 2.6. DENV-2 Quantitative Real Time-Polymerase Chain Reaction (qRT-PCR)

DENV-2 RNA was extracted from DENV-infected Vero cells using a QIAamp^®^ Viral RNA Mini kit (QIAGEN, Hilden, Germany). The reaction mixture was prepared according to the kit protocol (PrecisionPlus OneStep, Dorset, UK). Known titres of DENV, ranging from 1 × 10^2^ FFU to 1 × 10^6^ FFU, were used as reference standards. The qRT-PCR was performed using the DNA Engine Opticon^®^ system (Bio-Rad, Hercules, CA, USA) with the following thermal conditions: (i) reverse transcription at 50 °C for 30 min, (ii) initial denaturation at 95 °C for 10 min, (iii) 45 cycles of 95 °C for 15 s, 59 °C for 30 s, and 72 °C for 30 s. Primer set developed in-house and used for the amplification was as follows: forward primer (5′ CAA TAT GCT GAA ACG CGA GAG AAA 3′) and reverse primer (5′ AAG ACA TTG ATG GCT TTT GA 3′). A standard curve was plotted to determine the absolute quantities of viral RNA in the samples. The percentage of reduction of RNA copy numbers was calculated by comparing the tested concentrations of *Streptomyces* KSF 103 ME to the untreated virus control.

### 2.7. UHPLC-MS Analysis

Secondary metabolites profiling was performed using the Agilent 1290 Infinity UHPLC system coupled to Agilent 6520 Accurate-Mass Q-TOF mass spectrometer interfaced with a dual ESI source. The column used was Agilent Zorbax Eclipse XDB-C18 (2.1 × 150 mm, 3.5 μm). A temperature of 4 °C and 25 °C was maintained for the auto-sampler and column, respectively. The mobile phase (A) used was 0.1% formic acid solution in water, and acetonitrile, while 0.1% formic acid solution was the mobile phase (B). The flow rate of the mobile phase was kept at 0.5 mL/min. The *Streptomyces* KSF 103 ME (1.0 μL in HPLC grade methanol solvent) was injected and allowed to separate for 25 min, and an additional 5 min was utilized for post-run time. Nitrogen gas with a flow rate of 25 and 600 L/hour was used as a source of the nebulizing and drying gas, respectively, and the temperature was maintained at 350 °C. Analysis was performed with a capillary voltage of 3500 V while the fragmentation voltage was optimized to 125 V. Spectral data were analyzed using Agilent MassHunter Qualitative Analysis software and compared with known compounds based on similar molecular features. 

### 2.8. Molecular Docking

PyMol, AutoDock Vina 1.5.6, and Discovery Studio Visualizer 2.5 were utilized to prepare the docking files, perform the docking, and analyze the interaction and output. The aim of molecular docking is to give an early prediction of the ligand–receptor interaction. There were nine configurations produced via Autodock Vina and the first configuration with the highest binding affinity and stability was chosen for further analysis. The 2D structures of targeted ligands (compounds) were retrieved from PubChem (Figure 1), while the 3D structures of targeted receptors (DENV-2 proteins) were retrieved from Protein Data Bank (PDB) and PyMol (Figure 2). The active sites residue of each DENV-2 protein found to be interacting are shown in Table 1. In silico study to determine the binding affinity between the ligands and receptors was done as previously described [29]. 

### 2.9. Statistical Analysis

The value of 50% cytotoxic concentration (CC_50_) and IC_50_ of the extract were determined using Graph Pad Prism Version 5 (Graph Pad Software Inc., San Diego, CA, USA). The Selectivity Index (SI) value was determined as the ratio of CC_50_ to IC_50_ for the extract.

## 3. Results

### 3.1. In Vitro Cytotoxicity Assay and Screening for Inhibitory Activities of Streptomyces KSF 103 ME against DENV-2

The cytotoxicity of Streptomyces KSF 103 ME towards Vero cells was first determined. The MNTD and CC_50_ were recorded at 512.1 µg/mL and 790.3 µg/mL, respectively (Table 2). Using these values, the highest concentration of Streptomyces KSF 103 ME (500 µg/mL) used in this study was shown to be non-toxic to the cells (Figure 3A). The inhibitory activity of Streptomyces KSF 103 ME against DENV-2 was determined by treating the cells with Streptomyces KSF 103 ME at concentrations from 3.125 µg/mL to 50 µg/mL. Results shown in Table 2 suggest that Streptomyces KSF 103 ME have an IC_50_ and SI value of 20.3 µg/mL and 38.9 µg/mL, respectively, with a consistent reduction of DENV-2 RNA from the lowest concentration (3.125 µg/mL) to the highest concentration (50 µg/mL) (Figure 3B).

### 3.2. The Effects of Streptomyces KSF 103 ME Treatment at Different Stages of DENV-2 Replication Cycle

Streptomyces KSF 103 ME was evaluated against DENV-2 at different virus replication stages. The antiviral assays include anti-adsorption, anti-entry, anti-post-adsorption, prophylactic, and direct virucidal assays. The graph shows a consistent reduction of virus RNA copy numbers during the virus adsorption to the cells, from 12.5 µg/mL to 50 µg/mL, by 15%, 36%, and 46%, respectively, but exhibited no reduction in the lower concentrations (Figure 4A). The IC_50_ value was 16.59 µg/mL. During the virus internalization into the cells, a reduction in virus RNA copy numbers was consistently observed against the concentration gradient. Streptomyces KSF 103 ME inhibited DENV-2 entry by 1% and 19% at 3.125 µg/mL and 6.25 µg/mL, respectively, and fully inhibited (100%) DENV-2 entry at 12.5 µg/mL, 25 µg/mL, and 50 µg/mL with IC_50_ value of 6.94 µg/mL (Figure 4B). Once the virus entered the cells, the antiviral activity was observed only at the highest concentration (50 µg/mL) by 62% inhibition with an IC_50_ value of 51.60 µg/mL (Figure 4C). The antiviral activity of Streptomyces KSF 103 ME was also measured as a prophylactic treatment. There was no inhibition of DENV-2 at 3.125 µg/mL and 6.25 µg/mL, but a consistent inhibition was observed at 12.5 µg/mL, 25 µg/mL, and 50 µg/mL by 22%, 76%, and 90%, respectively, with an IC_50_ value of 16.56 µg/mL (Figure 4D). The extract was also tested as a direct virucidal treatment. No inhibition was recorded at low concentrations (3.125 µg/mL to 12.5 µg/mL), but antiviral activity was observed at 25 µg/mL and 50 µg/mL by 50% and 77%, respectively, with an IC_50_ value of 24.59 µg/mL (Figure 4E).

### 3.3. UHPLC-MS Analysis

UHPLC-MS analysis of Streptomyces KSF 103 ME was performed to identify potential compounds exhibiting antiviral properties against DENV. Negative and positive ion modes were used with the UHPLC-MS. Four different compounds were reported to exhibit potential antiviral properties based on similar molecular features of known compounds (Table 3).

### 3.4. Binding Affinity of Potential Antiviral Compounds in Streptomyces KSF 103 ME towards DENV-2 Proteins Using Molecular Docking

All four identified compounds were analyzed for binding affinity against DENV-2 proteins by molecular docking, as shown in Figure 5, Figure 6 and Figure 7. The top-scoring compounds, referred to as “ligands” (first ligand pose), with respective target DENV-2 proteins, are shown in Table 4. All ligands were shown to possess binding energy (kcal/mol) of −3.9 to −8.8 with the three DENV-2 proteins, namely 2FOM (NS2B/NS3 protease), 2j7u (NS5 polymerase), and 1OKE (envelope protein). Hypoxanthine showed binding energy of −5.1 kcal/mol, −4.9 kcal/mol, and −4.6 kcal/mol towards 2FOM, 2j7u, and 1OKE, respectively, while purine was shown to possess binding energy of −4.6 kcal/mol, −4.5 kcal/mol, and −4.0 kcal/mol with 2FOM, 2j7u, and 1OKE, respectively. Aminocaproic acid exhibited binding energy of −4.7 kcal/mol, −3.9 kcal/mol, and −4.3 kcal/mol towards 2FOM, 2j7u, and 1OKE, respectively. Meanwhile, 1α-fluoro-25-hydroxy-16,17,23,23,24,24-hexadehydrovitamin D3 (Vitamin D3) consistently showed the lowest binding energy against 2FOM, 2j7u, and 1OKE, at −8.8 kcal/mol, −7.9 kcal/mol, and −6.6 kcal/mol, respectively, compared to other ligands. Among the three DENV-2 proteins, vitamin D3 was shown to possess the highest binding affinity towards 2FOM with a binding energy score of −8.8 kcal/mol. 

## 4. Discussion

No effective treatment for dengue is presently available. While efforts to develop vaccines against the infection have resulted in at least two vaccines being approved, to date, no antivirals have successfully been developed. Several efforts to develop dengue antivirals from natural compounds are ongoing. Natural compounds, such as plant extracts, marine extracts, soil extracts, or secondary metabolites of microorganisms, are widely investigated for potential anti-infectives [30]. Several studies highlighted the beneficial antiviral properties of *Streptomyces* bioactive compounds in downregulating some pathogens [31,32,33,34]. For example, the methanolic fraction of the compound extracted from *Streptomyces* showed antiviral activity, which may interfere with the adsorption and/or penetration of the virus particles into the cells [35]. The two bioactive compounds from *Streptomyces* sp., WS-30581 A and WS-30581 B, were investigated and shown to hinder the early stage of EV71 replication [31]. There are also other studies highlighting the discoveries of *Streptomyces* bioactive compounds exhibiting antiviral properties against other viruses such as herpes simplex virus 1 (HSV-1), influenza A virus (IAV), tobacco mosaic virus (TMV), and varicella zoster virus (VZV) [32,33,34]. In this study, *Streptomyces* KSF 103 ME was derived from soil sediments of a primary forest in Malaysia. The extract was then tested via antiviral assays and characterized for probable antiviral properties against DENV-2 replication. 

The CC_50_ and MNTD of *Streptomyces* KSF 103 ME were first determined by in vitro cytotoxicity assay. The CC_50_ described the concentration that resulted in the reduction in cell viability by 50% and any concentrations more than CC_50_ were considered toxic, while the MNTD was indicative of cell viability at 90% for infection with DENV-2 to occur. Results from this study suggested that *Streptomyces* KSF 103 ME was non-toxic to the cells as it can be tolerated to as high as 512.1 µg/mL. In the study, the concentrations of *Streptomyces* KSF 103 ME were fixed at 50 µg/mL since the extract at 50 µg/mL reduced DENV-2 RNA copy numbers by 80% with 100% cell viability. *Streptomyces* KSF 103 ME showed a consistent inhibition of DENV-2 in a dose-dependent manner. *Streptomyces* KSF 103 ME was further tested to see the efficacy against DENV-2 at the different replication stages.

Our result suggested *Streptomyces* KSF 103 ME is most effective in inhibiting DENV-2 during entry. The mechanisms of the inhibition at this stage are still unknown. Inhibition of DENV replication at the entry stage could hinder virus entry into host cells and stop virus replication. This will potentially reduce the level of viremia and prevent the progression of severe dengue. DENV initially infected dendritic cells and Langerhans cells before infecting other surrounding organs such as the liver, spleen, and kidney tissues, and eventually accelerated to severe dengue, which manifests severe plasma leakage [36]. Viremia level was shown to be higher in secondary dengue infection which may prolong the disease and increase the chances of manifesting severe dengue symptoms [5,11]. Thus, inhibiting viral entry into the host cells could potentially prevent further viral replication and subsequently curb dengue transmission.

Based on screening via UHPLC-MS, four out of eighty-six compounds were identified from *Streptomyces* KSF 103 ME that may have potential antiviral properties, namely hypoxanthine, vitamin D3, purine, and aminocaproic acid. Hypoxanthine was previously reported to show inhibitory activity against flaviviruses, alphaviruses, arenaviruses, and rhabdoviruses [37]. Vitamin D3, among those that existed in *Streptomyces* KSF 103 ME, exhibited antiviral activity against the hepatitis C virus (HCV), hepatitis B virus (HBV), and HSV-1 [38,39,40,41]. Purine, a nucleoside analogue, and its derivatives have been widely studied against different viruses such as HIV, HSV-1, HSV-2, VZV, cytomegalovirus, and HBV [37,42,43,44,45]. Finally, aminocaproic acid was also reported in earlier studies to have inhibitory activity against the respiratory syncytial virus and influenza virus [46,47]. The presence of these four compounds may exert antiviral activities against DENV-2, and this was evaluated in the present study. 

Molecular docking studies were performed to identify the binding affinity of these compounds against three important DENV-2 proteins. All compounds were shown to bind to targeted DENV-2 proteins, and the highest affinity binding was towards 2FOM (NS2B/NS3 protease). NS2B initially wraps itself around NS3, forming an NS2B/NS3 complex, which plays a vital role in virus maturation and RNA replication. *Streptomyces* KSF 103 ME may inhibit DENV-2 replication by interrupting the activity of NS2B/NS3 protease, suggesting that it is a promising target for anti-DENV-2 drug development [14]. In addition to 2FOM, these compounds also possessed binding affinity towards 2j7u (NS5 polymerase) and 1OKE (envelope protein). NS5 polymerase is involved in virus replication by the capping of RNA and host cell gene regulation [16], while envelope protein is the mediator for viral entry and membrane fusion with a receptor on the host cell [18]. The binding of these identified compounds with NS5 polymerase and envelope protein may hinder viral replication and prevent the further spread of dengue infection. Though all compounds were shown to bind to all DENV-2 proteins, vitamin D3 possessed the highest binding affinity towards 2FOM. Little is presently known about *Streptomyces* KSF 103 ME in inhibiting DENV-2, a strong potential interaction of DENV-2 proteins with the active sites of these compounds, identified in silico, warrants further investigation [48].

Our findings showed that *Streptomyces* KSF 103 ME has promising antiviral properties against DENV-2, especially as an inhibitor during virus entry into the host cells. This is an important contribution to the field as it suggests that the extract has the potential to reduce the level of viremia, which may prevent the progression of severe dengue. By highlighting specific stages of viral replication targeted by the extract, this study provides valuable insights for developing antivirals against dengue. The inclusion of known dengue antivirals would have strengthened the study by allowing a direct comparison between *Streptomyces* KSF 103 ME and other antivirals. There are, however, no antivirals available for dengue, which is the limitation of this study. While this study focused on the in vitro evaluation of the extract, further investigations to evaluate its safety and efficacy in the in vivo models are required. Future research should also explore the potential mechanisms and pharmacokinetic properties of the active compounds in *Streptomyces* KSF 103 ME in inhibiting DENV-2 replication.

## 5. Conclusions

The *Streptomyces* strain KSF 103 methanolic extract exhibited inhibitory properties against DENV-2 at different stages of the virus replication cycle, especially during the entry stage. Inhibiting the entry of the virus into the cells and causing subsequent infection may be the key target in developing antivirals against the disease. The extract with its minimal cytotoxicity and robust antiviral properties at low concentrations had a potential for further development as an antiviral drug against dengue. These findings contribute to the field of antiviral drug discovery as they highlight the potential of natural products derived from *Streptomyces* strains as a source of novel antiviral agents. Further investigations, however, should explore the effectiveness of the extract against other serotypes of the dengue virus and the potential antiviral mechanisms of the *Streptomyces* strain KSF 103 methanolic extract. Additionally, while our study focused on the in vitro antiviral properties of the extract, future studies should assess its efficacy in relevant animal models and eventually progress to clinical trials to evaluate its safety and efficacy.

## Figures and Tables

**Figure 1 viruses-15-01773-f001:**
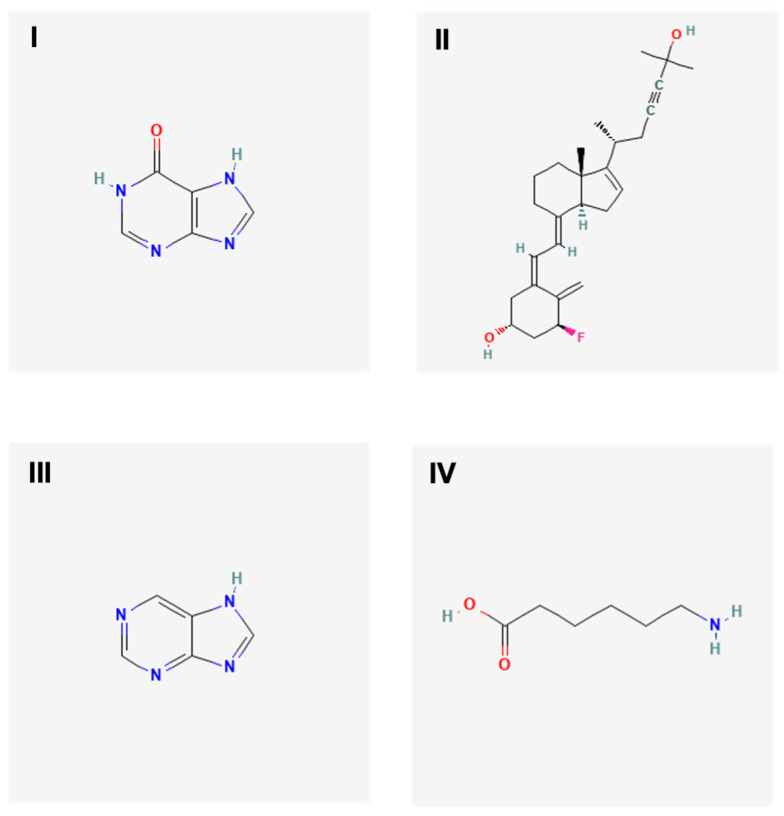
The 2D molecular structures of targeted ligands. Targeted ligands were hypoxanthine (**I**), 1α-fluoro-25-hydroxy-16,17,23,23,24,24-hexadehydrovitamin D3 (**II**), purine (**III**) and aminocaproic acid (**IV**).

**Figure 2 viruses-15-01773-f002:**
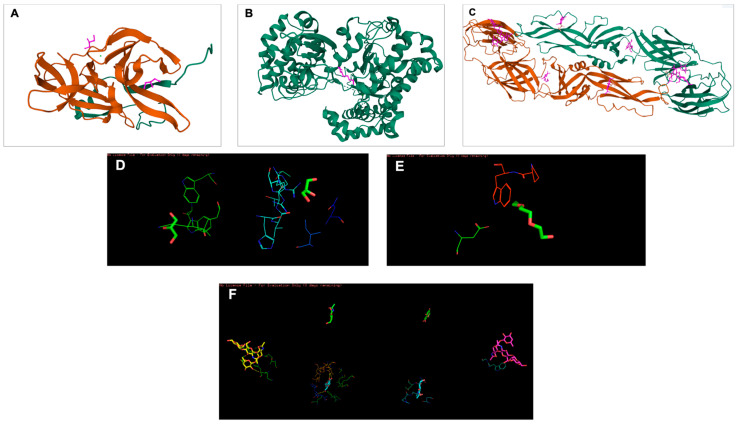
The 3D structures of targeted DENV-2 proteins derived from the PDB server. (**A**) Structure of NS2B/NS3 protease (ID: 2FOM). Orange: NS3pro, Green: NS2B. Two existing ligands were found interacting in the structure. (**B**) NS5 polymerase (ID:2j7u). Only one existing ligand was found in the structure. (**C**) E protein (ID: 1OKE). The structure consists of 6 existing ligands. (**D**–**F**) Structures of 2FOM, 2j7u and 1OKE visualized in PyMol, respectively.

**Figure 3 viruses-15-01773-f003:**
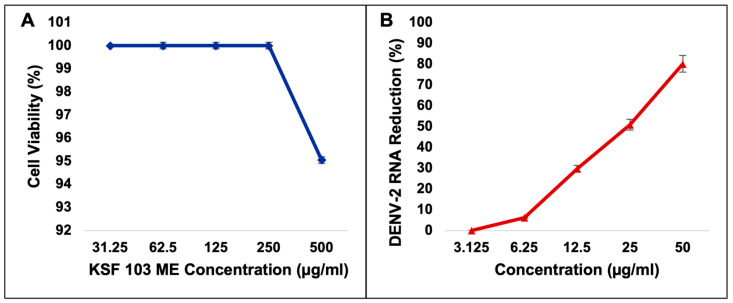
Cytotoxicity of Streptomyces KSF 103 ME against Vero cells (**A**) and screening for inhibitory activities of Streptomyces KSF 103 ME against DENV-2 (**B**). The extract was non-toxic to Vero cells even at the highest concentration, and the extract showed a consistent reduction in DENV-2 RNA against the concentration gradient.

**Figure 4 viruses-15-01773-f004:**
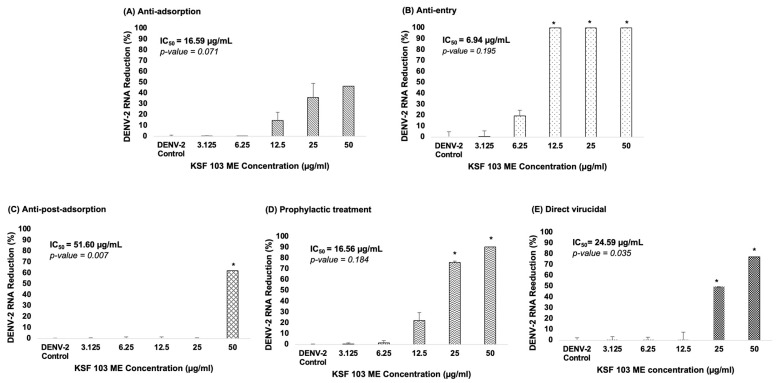
(**A**–**E**) Dose-dependent antiviral activity of Streptomyces KSF 103 ME at different stages of the DENV-2 replication cycle. The DENV-2 infected cells were treated with the shown concentration of Streptomyces KSF 103 ME in duplicate and the graph was presented with error bars representing standard deviation. DENV-2 copy numbers were measured via qRT-PCR and the reduction percentage was calculated as shown (* Reduction ≥ 50%).

**Figure 5 viruses-15-01773-f005:**
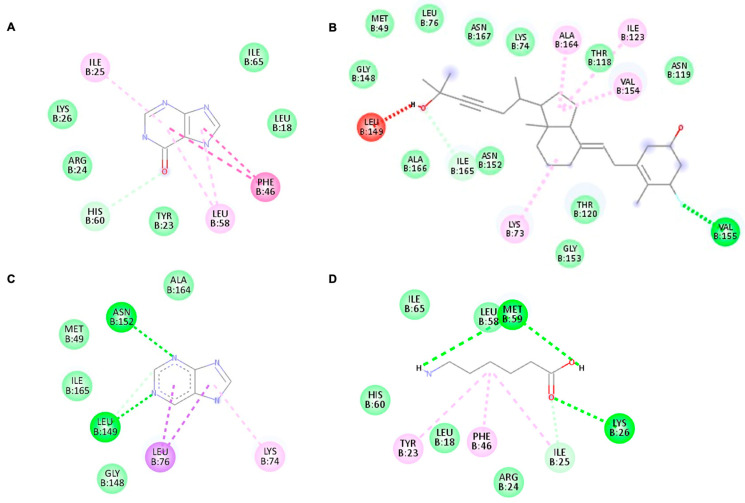
The highest binding affinity (first ligand pose) of each ligand with NS2B/NS3 protease (ID:2FOM) generated using Discovery Studio Visualizer. (**A**–**D**) The interaction of hypoxanthine, vitamin D3, purine, and aminocaproic acid with 2FOM, respectively.

**Figure 6 viruses-15-01773-f006:**
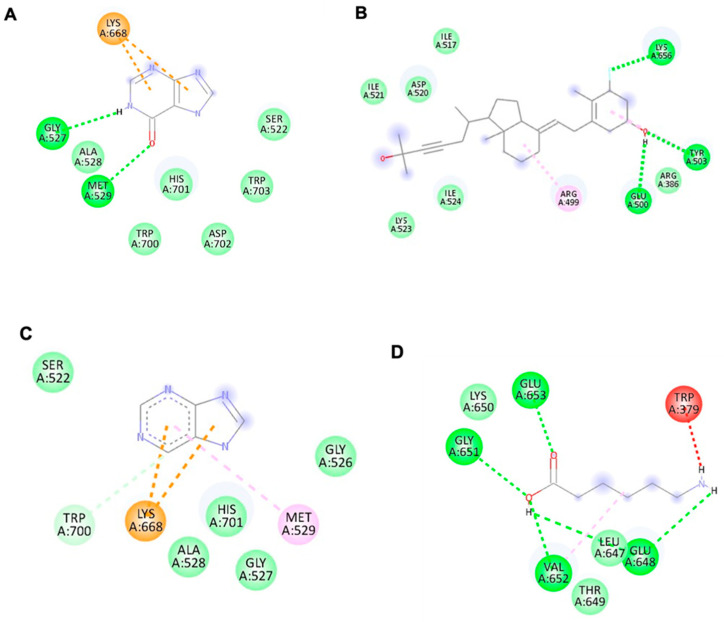
The interaction of each ligand with NS5 polymerase (ID:2j7u). (**A**–**D**) The interaction of hypoxanthine, vitamin D3, purine, and aminocaproic acid with 2j7u, respectively.

**Figure 7 viruses-15-01773-f007:**
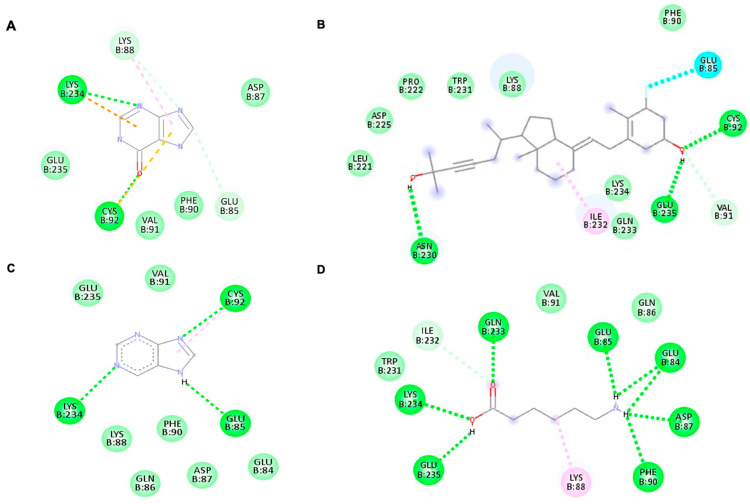
The interaction of each ligand with envelope protein (ID:1OKE) visualized using Discovery Studio Visualizer. (**A**–**D**) The interaction of hypoxanthine, vitamin D, purine, and aminocaproic acid with 1OKE, respectively.

**Table 1 viruses-15-01773-t001:** Predicted active site residues for each targeted DENV-2 protein obtained using PyMol.

Targeted DENV-2 Protein	Active Site Residues
NS2B/NS3 protease (ID:2FOM)	TRP-69, LYS-74, LEU-76, TRP-83
NS5 polymerase (ID: 2j7u)	ASP-520, PRO-822, TRP-823
Envelope protein (ID: 1OKE)	ASN-67, GLU-84, ARG-89, PHE-90, MET-118

**Table 2 viruses-15-01773-t002:** The CC_50_, MNTD, IC_50_, and SI values of Streptomyces KSF 103 ME.

Extract	CC_50_ (µg/mL)	MNTD (µg/mL)	IC_50_ (µg/mL)	SI Value
*Streptomyces* strain KSF 103 methanolic extract	790.3	512.1	20.3	38.9

**Table 3 viruses-15-01773-t003:** Composition of Streptomyces KSF 103 ME using UHPLC-MS.

No	Proposed Compounds	Chemical Formula
1	Hypoxanthine [12,13,14]	C_5_H_4_N_4_O
2	1α-fluoro-25-hydroxy-16,17,23,23,24,24-hexadehydrovitamin D3/1α-fluoro-25-hydroxy-16,17,23,23,24,24-hexadehydrocholecalciferol [15,16,17,18]	C_27_H_37_FO_2_
3	Purine [12,14,19,20,21]	C_5_H_4_N_4_
4	Aminocaproic acid [22,23]	C_6_H_13_NO_2_

**Table 4 viruses-15-01773-t004:** Top binding affinity of each ligand with respective target DENV-2 proteins.

Protein	Ligand	Binding Energy (kcal/mol)
2FOM	Hypoxanthine	−5.1
	Vitamin D3	−8.8
	Purine	−4.6
	Aminocaproic acid	−4.7
2j7u	Hypoxanthine	−4.9
	Vitamin D3	−7.9
	Purine	−4.5
	Aminocaproic acid	−3.9
1OKE	Hypoxanthine	−4.6
	Vitamin D3	−6.6
	Purine	−4.0
	Aminocaproic acid	−4.3

## Data Availability

Not applicable.

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
