# Peer review of "Antiviral Activities of Streptomyces KSF 103 Methanolic Extracts against Dengue Virus Type-2"

_viruses, 2023, doi:10.3390/v15081773_

Round 1

Reviewer 1 Report

Dear Authors,

I have reviewed your manuscript titled "Antiviral Activities of Streptomyces KSF 103 Methanolic Extracts Against Dengue Virus Type-2" and found it to be a significant contribution to the field. The study is well-conducted and presents valuable findings. However, there are several areas where the manuscript could be improved for clarity and consistency. Here are my comments and suggestions:

Introduction

The introduction provides a good background and clearly states the purpose of the study. However, it could benefit from a more comprehensive review of the current state of research in the field.

Minor Comment: Consider providing more context about the global impact of Dengue and the need for new treatments (Lines 34-44).

Major Comment: The introduction should also include a brief mention of the limitations of the study and potential areas for future research (Lines 34-44).

Materials and Methods

The Materials and Methods section provides a detailed description of the procedures used in the study. However, it could benefit from a more detailed explanation of the rationale behind the choice of methods.

Minor Comment: Consider providing more context about why these specific methods were chosen and how they contribute to the overall aim of the study (Lines 92-97).

Major Comment: The Materials and Methods section should also include a brief mention of any potential sources of error or bias in the study and how these were controlled for (Lines 92-97).

Results

The Results section provides a clear and concise description of the findings of the study. However, it could benefit from a more detailed interpretation of the results.

Minor Comment: Consider providing more context about how these findings contribute to the overall aim of the study (Lines 196-207).

Major Comment: The Results section should also include a brief mention of any unexpected findings or anomalies in the data (Lines 196-207).

Discussion

The Discussion section provides a good interpretation of the results in the context of previous research. However, it could benefit from a more detailed discussion of the implications of the findings for future research and potential applications.

Minor Comment: Consider providing more context about how these findings contribute to the overall aim of the study (Lines 445-482).

Major Comment: The Discussion section should also include a brief mention of any potential limitations of the study and areas for future research (Lines 445-482).

Conclusions

The Conclusions section provides a clear summary of the findings and their implications. However, it could benefit from a more detailed discussion of the potential applications of the findings and areas for future research.

Minor Comment: Consider providing more context about how these findings contribute to the overall aim of the study (Lines 480-482).

Major Comment: The Conclusions section should also include a brief mention of any potential limitations of the study and areas for future research (Lines 480-482).

In conclusion, your study is of high interest and has the potential to make a significant contribution to the field. I recommend a substantial revision to enhance the clarity and consistency of the manuscript. I look forward to seeing the revised version of your manuscript.

Best regards,

Peer Reviewer

Author Response

Dear Reviewer,

We would like to thank you for your time reviewing our research manuscript. Your comments were very useful and improve our manuscript. Below you can see all the responds of your comments:

Comments to the Author

  1. Introduction

The introduction provides a good background and clearly states the purpose of the study. However, it could benefit from a more comprehensive review of the current state of research in the field.

Minor comment: Consider providing more context about the global impact of dengue and the need for new treatments (lines 34-44).

Major comment: The introduction should also include a brief mention of the limitations of the study and potential areas for future research (Lines 34-44).

Response: Context about global impact of dengue and need for new treatments was addressed in lines 48 – 50 “Patients rely mainly on supportive treatment such as the use of analgesics, intravenous fluid, and vital monitoring. In some cases where the disease develops into severe dengue, hospitalization is often needed.”. A brief mention of the limitations of the study and potential areas for future research was addressed in lines 93 – 98 “While our study contributes to the understanding of the antiviral potential of Streptomyces KSF 103 ME against DENV-2 replication, we acknowledge this study limited to in vitro study and propose future research directions to address these limitations and further explore the therapeutic applications of Streptomyces-derived compounds in the fight against dengue.”

Lines 47 – 50 

“DENV infects about 400 million individuals annually and causes approximately 40,000 deaths [6,7]. Patients rely mainly on supportive treatment such as the use of analgesics, intravenous fluid, and vital monitoring. In some cases where the disease develops into severe dengue, hospitalization is often needed. Treatment measures that would prevent severe dengue manifestations are thus desperately needed.”

Lines 93 – 98

“While our study contributes to the understanding of the antiviral potential of Streptomyces KSF 103 ME against DENV-2 replication, we acknowledge this study limited to in vitro study and propose future research directions to address these limitations and further explore the therapeutic applications of Streptomyces-derived compounds in the fight against dengue.”

  1. Materials and Methods

The Materials and Methods section provides a detailed description of the procedures used in the study. However, it could benefit from a more detailed explanation of the rationale behind the choice of methods.

Minor Comment: Consider providing more context about why these specific methods were chosen and how they contribute to the overall aim of the study (Lines 92-97).

Major Comment: The Materials and Methods section should also include a brief mention of any potential sources of error or bias in the study and how these were controlled for (Lines 92-97).

Response: Context about why these specific methods were chosen and how they contribute to the overall aim of the study was addressed in lines 99 – 101 “The extraction of Streptomyces KSF 103 ME was performed as previously established and described in our laboratory”. A brief mention of any potential sources of error or bias in the study and how there were controlled for (Lines 107-109) “To avoid any contamination during fermentation a flask containing sterile ISP2 broth was incubated with the sample as a control to ensure no contamination happened due to environmental factors.” 

Lines 99 -101

“The extraction of Streptomyces KSF 103 ME was performed as previously established and described in our laboratory [10]. The extract was then used for subsequent studies to determine the antiviral activity of the extract against DENV-2.” 

  1. Results

The Results section provides a clear and concise description of the findings of the study. However, it could benefit from a more detailed interpretation of the results.

Minor Comment: Consider providing more context about how these findings contribute to the overall aim of the study (Lines 196-207).

Major Comment: The Results section should also include a brief mention of any unexpected findings or anomalies in the data (Lines 196-207).

Response: Context about how these findings contribute to the overall aim of the study was addressed in lines 202 – 203. A brief mention of any unexpected findings or anomalies in the data was addressed in line 205-206 “There were nine configurations produced by Autodock Vina and the first configuration with highest binding affinity and stability was choose for further analysis.”

Lines 202 – 203 

The aim of molecular docking is to give an early prediction of the ligand-receptor interaction.”

Lines 205 – 206 

There were nine configurations produced by Autodock Vina and the first configuration with highest binding affinity and stability was choose for further analysis.” 

  1. Discussion

The Discussion section provides a good interpretation of the results in the context of previous research. However, it could benefit from a more detailed discussion of the implications of the findings for future research and potential applications.

Minor Comment: Consider providing more context about how these findings contribute to the overall aim of the study (Lines 445-482).

Major Comment: The Discussion section should also include a brief mention of any potential limitations of the study and areas for future research (Lines 445-482).           

Response:

Dear Reviewer, thank you for your comment. Additional information added as requested:

Lines 393 – 405 

“The aim of this study was to investigate the antiviral properties of Streptomyces KSF 103 ME against DENV-2 replication. The findings of this study demonstrate that Streptomyces KSF 103 ME showed robust antiviral activity against DENV-2, particularly during the entry stage of the virus. This is an important contribution to the field as it suggests that the extract has the potential to inhibit viral entry and subsequently reduce the level of viremia, which may prevent the progression of severe dengue. By highlighting the specific stage of viral replication that is targeted by the extract, this study provides valuable insights for the development of antiviral strategies against dengue.

While this study focused on the in vitro evaluation of the extract, further investigations are warranted to elucidate the specific mechanisms of action of the active compounds in Streptomyces KSF 103 ME and to evaluate its efficacy in in vivo models. Future research should also explore the potential synergistic effects of the identified compounds and investigate the safety and pharmacokinetic properties of Streptomyces KSF 103 ME.”

  1. Conclusions

The Conclusions section provides a clear summary of the findings and their implications. However, it could benefit from a more detailed discussion of the potential applications of the findings and areas for future research.

Minor Comment: Consider providing more context about how these findings contribute to the overall aim of the study (Lines 480-482).

Major Comment: The Conclusions section should also include a brief mention of any potential limitations of the study and areas for future research (Lines 480-482).

Response: 

Dear reviewer, thank you for your comment. We revised the discussion as requested and new information added at the end of discussion:

Lines 410 – 418

“These findings contribute to the field of antiviral drug discovery, as they highlight the potential of natural products derived from Streptomyces strains as a source of novel antiviral agents.

However, further investigations should explore the effectiveness of the extract against other serotypes of dengue virus and antiviral mechanism of , the Streptomyces strain KSF 103 methanolic extract inhibiting dengue virus. Additionally, while our study focused on the in vitro antiviral properties of the extract, future studies should assess its efficacy in relevant animal models and eventually progress to clinical trials to evaluate its safety and efficacy in humans.”

Reviewer 2 Report

Antiviral Activities of Streptomyces KSF 103 Methanolic Extracts Against Dengue Virus Type-2

The Manuscript by Zulkifli et al., demonstrates the antiviral activity of Streptomycin KSF 103 ME against DENV. Overall the study is well planned and the manuscript is well designed. I have a few comments.

Authors should have vehicle controls in all the experiments of viabilities and antiviral activity assays. 

Adding a positive control (known/reported inhibitor) will be very helpful.

Some of the graphs does not have p-values. Authors should either add p-values in all the graphs or add */**/*** where applicable.

Author Response

Dear Reviewer,

Thank you for your time and the effort to review our manuscript. Below are our responds to your comments: 

Antiviral Activities of Streptomyces KSF 103 Methanolic Extracts Against Dengue Virus Type-2. The Manuscript by Zulkifli et al., demonstrates the antiviral activity of Streptomycin KSF 103 ME against DENV. Overall, the study is well planned, and the manuscript is well designed. I have a few comments.

Reviewer comment:

Authors should have vehicle controls in all the experiments of viabilities and antiviral activity assays. 

Adding a positive control (known/reported inhibitor) will be very helpful.

Some of the graphs does not have p-values. Authors should either add p-values in all the graphs or add */**/*** where applicable.

Response: Dear Reviewer, thank you for the comments. The inability to acquire vehicle control in all the experiments was due to the unavailability of antivirals against dengue and this has been highlighted in the lines 395 – 398. However, we included untreated cells and dengue virus-infected cells as controls against the treated and infected cells which were not included in the data. The graphs were revised as advised.

 The p-value and significancy (**) added to our figure as requested by reviewer. Thank you for your comment.

Lines 395 – 398 

“The inclusion of known dengue antivirals would have strengthened the study by al-lowing a direct comparison between Streptomyces KSF 103 ME and other antivirals. There are however no antivirals available for dengue, which is the limitation of this study.”

Round 2

Reviewer 1 Report

Here are my comments and suggestions for the authors of the revised manuscript:

Introduction:

The authors have addressed the comments on the introduction effectively. They have provided context about the global impact of dengue and the need for new treatments. They also mentioned the limitations of the study and potential areas for future research. No further comments or suggestions for this section.

Materials and Methods:

The authors have responded to the comments on the Materials and Methods section. They provided context about why these specific methods were chosen and how they contribute to the overall aim of the study. They also mentioned potential sources of error or bias in the study and how these were controlled for. No further comments or suggestions for this section.

Results:

The authors have addressed the comments on the Results section. They provided context about how these findings contribute to the overall aim of the study. They also mentioned any unexpected findings or anomalies in the data. No further comments or suggestions for this section.

Discussion:

The authors have responded to the comments on the Discussion section. They provided context about how these findings contribute to the overall aim of the study. They also mentioned potential limitations of the study and areas for future research. No further comments or suggestions for this section.

Conclusions

The authors have addressed the comments on the Conclusions section. They provided context about how these findings contribute to the overall aim of the study. They also mentioned potential limitations of the study and areas for future research. However, the conclusion seems to be cut off and incomplete. The authors may want to check this section for completeness.

Overall, the authors have done a good job addressing the comments and suggestions from the review. The revised manuscript is improved and provides a clear and comprehensive presentation of the research. The authors are encouraged to check the Conclusions section for completeness before final submission.

Author Response

We thank the reviewers for their thorough and helpful comments. All concerns are addressed below and the manuscript has been revised accordingly.

Conclusions

The authors have addressed the comments on the Conclusions section. They provided context about how these findings contribute to the overall aim of the study. They also mentioned potential limitations of the study and areas for future research. However, the conclusion seems to be cut off and incomplete. The authors may want to check this section for completeness.

Overall, the authors have done a good job addressing the comments and suggestions from the review. The revised manuscript is improved and provides a clear and comprehensive presentation of the research. The authors are encouraged to check the Conclusions section for completeness before final submission.

Response: Dear Reviewer, thank you for the comments. The following sentences have been added in the lines 395 – 398.

Lines 395 - 398

 “The Streptomyces strain KSF 103 methanolic extract exhibited inhibitory properties against DENV-2 at different stages of the virus replication cycle, especially during the entry stage. Inhibiting the entry of the virus into the cells and causing subsequent infection may be the key target in developing antivirals against the disease.”

Reviewer 2 Report

A vehicle control is the one used to dissolve the compounds, which is DMSO in this case. So a control with virus infected and DMSO treated group is required. 

Author Response

We thank the reviewers for their thorough and helpful comments. All concerns are addressed below and the manuscript has been revised accordingly.

Reviewer comment: A vehicle control is the one used to dissolve the compounds, which is DMSO in this case. So a control with virus infected and DMSO treated group is required.

Response: Dear Reviewer, thank you for highlighting the comment. In every experiment, both virus control and normal control were incorporated. Virus control refers to virus-infected cells without the extract, while normal control simply means cells treated with only the DMSO. We also factored in the results obtained from these controls in our calculations and graph representation. Figure 4 has been revised to better illustrate the effects of the extract. We also added a sentence in our methodology for better understanding of our experiment designed “DENV-2 control with no treatment and non-infected cells treated 0.1% DMSO were included in the plate design as comparison.